# Revealing myopathy spectrum: integrating transcriptional and clinical features of human skeletal muscles with varying health conditions

Huahua Zhong [1] ✉, Veronica Sian [2], Mridul Johari [3,4], Shintaro Katayama [3,5], Ali Oghabian [3,6], Per Harald Jonson [3], Peter Hackman[3], Marco Savarese[3] & Bjarne Udd[3,7]

Myopathy refers to a large group of heterogeneous, rare muscle diseases. Bulk RNA-sequencing has been utilized for the diagnosis and research of these diseases for many years. However, the existing valuable sequencing data often lack integration and clinical interpretation. In this study, we integrated bulk RNA-sequencing data from 1221 human skeletal muscles (292 with myopathies, 929 controls) from both databases and our local samples. By applying a method similar to single-cell analysis, we revealed a general spectrum of muscle diseases, ranging from healthy to mild disease, moderate muscle wasting, and severe muscle disease. This spectrum was further partly validated in three specific myopathies (97 muscles) through clinical features including trinucleotide repeat expansion, magnetic resonance imaging fat fraction, pathology, and clinical severity scores. This spectrum helped us identify 234 genuinely healthy muscles as unprecedented controls, providing a new perspective for deciphering the hallmark genes and pathways among different myopathies. The newly identified featured genes of general myopathy, inclusion body myositis, and titinopathy were highly expressed in our local muscles, as validated by quantitative polymerase chain reaction.

Myopathy is a general term that refers to a large group of diseases primarily affecting the skeletal muscles. These conditions can be categorized into inherited or acquired forms, according to their aetiology. Myopathies exhibit heterogeneous phenotypes, including weakness, abnormal gait, muscle pain, difficulty swallowing, contractures, and systemic impairments, among others[1–3]. One common feature of myopathies is a large spectrum of the degree of severity, similar to other progressive diseases, which includes asymptomatic, mild-to-moderate, and severe stages[4–7]. However, this spectrum is usually more based on clinical observation than on well-established objective findings.

RNA-sequencing for bulk skeletal muscles has been utilized in the diagnosis and research of muscle diseases (e.g., for ectopic splicing and molecular mechanism investigation)[8–10]. It offers a sensitive perspective to understand the ongoing molecular activities in the muscle. Numerous studies have deposited their transcriptional data in various online databases, and this data is of significant value for integration, especially considering the rarity of most myopathies. However, these isolated datasets are somewhat prone to various biases from small sample size, selection of control materials, sequencing methods, different read length, etc[11].

Furthermore, with the emergence of more advanced and sophisticated technologies developed for myopathies in the past decade, deep phenotyping (including quantitative MRI, muscle biopsy evaluation, CTG expansion size in myotonic dystrophy) provides a multi-dimensional evaluation to depict the muscle deterioration process. Correlating these muscle-specific features with their genetic data can assist in characterizing and deciphering myopathies.

In this study, we integrated transcriptional data from 1221 human skeletal muscles obtained from both online databases and our local patients,

[1]Department of Neurology, Huashan Rare Disease Center, Huashan Hospital, Fudan University, Shanghai, China. [2]Department of Precision Medicine, "Luigi Vanvitelli" University of Campania, Via L. De Crecchio 7, Naples, Italy. [3]Department of Medical and Clinical Genetics, Folkhälsan Research Center, Medicum, University of Helsinki, Helsinki, Finland. [4]Harry Perkins Institute of Medical Research, Centre for Medical Research, University of Western Australia, Nedlands, WA, Australia. [5]Department of Biosciences and Nutrition, Karolinska Institutet, Huddinge, Sweden. [6]Research Program for Clinical and Molecular Metabolism, Faculty of Medicine, University of Helsinki, Helsinki, Finland. [7]Tampere Neuromuscular Center, University Hospital, Tampere, Finland. ✉e-mail: huatwofold@outlook.com

ultimately identifying a general spectrum separating normal and myopathy-affected muscles. In contrast to the traditional approach of focusing primarily on diseased samples, we reversed the perspective, emphasizing the control samples to characterize this spectrum validated using clinical features from different sources. We offered a novel perspective by using genuinely healthy muscles as an unprecedented control reference, aiming to identify both common pathways and specific features of the studied myopathies.

## Methods

### Data source and participant selection

This is a retrospective integrative analysis (Fig. 1). The data sources include 803 muscles from the GTEx Consortium (dbGaP Accession phs000424.v8.p2)[12], 291 muscles from the GEO database (GSE115650[13], GSE140261[14], GSE175861[15], GSE184951[16], GSE201255[17], GSE202745[18]), and 127 muscles from Helsinki (39 of which have also been reported as GSE151757[19]). The ethics approval of using local muscles (195/13/03/00/11) was approved by HUS (Helsingin Uudenmaan Sairaanhoitopiiri) and informed consent was obtained from each subject. All ethical regulations relevant to human research participants were followed. The inclusion and exclusion criteria for participant selection were as follows: (1) only human skeletal muscle tissue was included (no cell lines or organoids); (2) bulk-RNA sequencing was performed using high-throughput techniques (no chip arrays or single-cell data); (3) datasets were preserved in raw count format (those shared in transformed count format were excluded).

A total of 23 phenotypes were used to annotate participants, encompassing 15 myopathies and 8 different controls. The diagnoses of myopathy patients were based on their genetic, pathology, electromyography, and radiological evidence. Patients with unspecific genetic causes were identified as "Myopathy (unsolved)". Considering the rarity of myopathy diseases (around 4 per 100,000 people globally), we presumptively categorized all GTEx muscles as not involved with myopathy[20]. In total, 291 muscles were sequenced using the total RNA method, while 930 were sequenced using the mRNA method (292

myopathies, 929 controls). The meta data for each muscle in the integration dataset is listed in Supplementary Data 1.

### Preprocessing and integration analysis

The integrated raw data counts were adjusted using a negative-binomial-regression-based batch effect adjustment tool, ComBat-seq (for bulk RNA-seq count data), and a normalization algorithm, Trimmed Mean of M-values (TMM) (https://gitlab.com/georgy.m/conorm), which is presumably better for between-sample comparisons[21,22]. Different gene sets across all 1221 samples were initially intersected, resulting in 16,953 candidate genes. To minimize the impact of genes with low expression, we applied a straightforward yet stringent filtering rule to each sample: counts for muscle-specific genes must exceed 0 in all samples. Following this criterion, 9231 genes were selected. The visualization of the integrated dataset was conducted using a single-cell analysis pipeline (Scanpy), which includes principal component analysis (PCA) and uniform manifold approximation and projection (UMAP) analyses[23]. The data and code used for this study are publicly available at GitHub (https://github.com/Hirririririr/Myopathy_spectrum).

### Mapping the UMAP spectrum with clinical features from different myopathies

The myopathy spectrum order was validated using in-silico analysis and clinical data sourced from the supplementary data of original studies. Pseudo-time analysis and trajectory prediction (PAGA) were used to in-silico predict the transformation ranks from healthy muscles to myopathy-affected muscles[24]. Clinical features including *DMPK* CTG repeat number (peripheral blood), Mercuri score from conventional MRI (cMRI), fat fraction from quantitative MRI (qMRI), pathology and inflammation scores, clinical severity score, 10-meter walk test, and 6-minute walk test were mapped onto the integral UMAP result to validate the myopathy spectrum. Note that only sequenced muscles with available clinical information were analyzed. Given that there are no linear relationships among all muscles in the UMAP plot, for each myopathy, we categorized the muscles into three categories by their UMAP X location and tested whether an

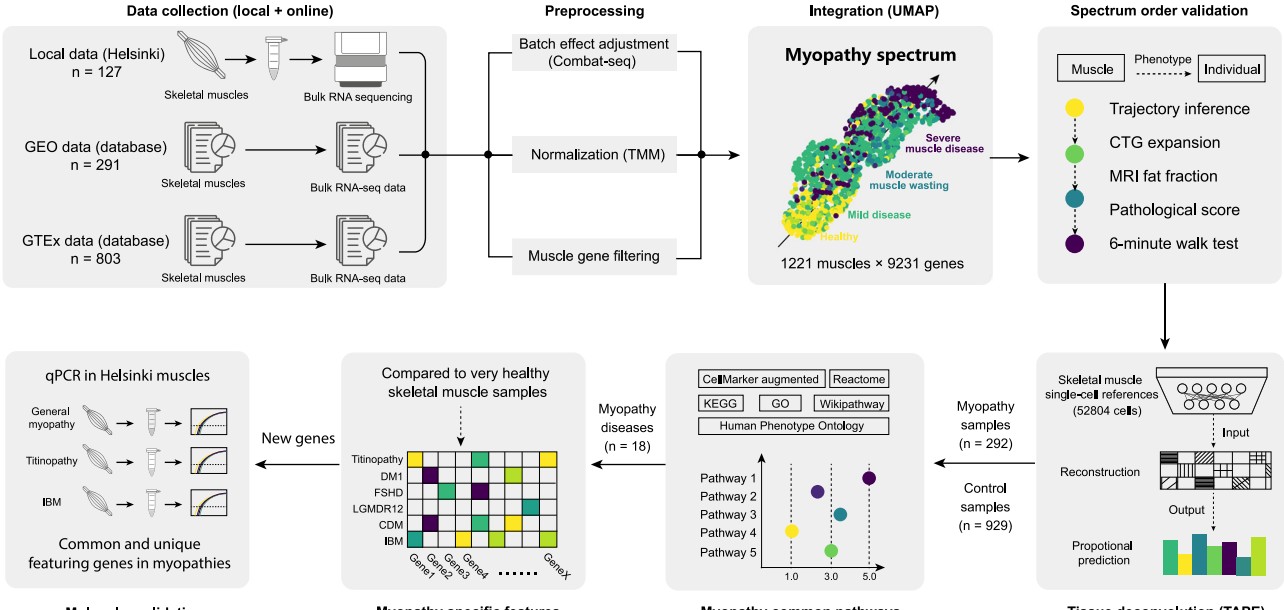

**Fig. 1 | The workflow.** Human skeletal muscle bulk-RNA-seq data from three sources (GTEx database, GEO database, and Helsinki) were integrated into a combined dataset (1221 muscles × 9231 genes). A spectrum order can be observed in this integrated dataset: Healthy→Mild disease→Moderate muscle wasting →Severe muscle disease. Different clinical features were mapped to the transcriptional data to validate this spectrum order. Tissue deconvolution was performed using skeletal

muscle single-cell datasets as references, allowing us to infer the cell type composition in myopathy muscles. By utilizing genuinely healthy muscles from the GTEx database as controls, we conducted differential expression analyses on general myopathy and six distinct myopathies to explore shared and unique featured genes and pathways among them. Finally, these newly identified featured genes were validated in the Helsinki muscle samples using qPCR.

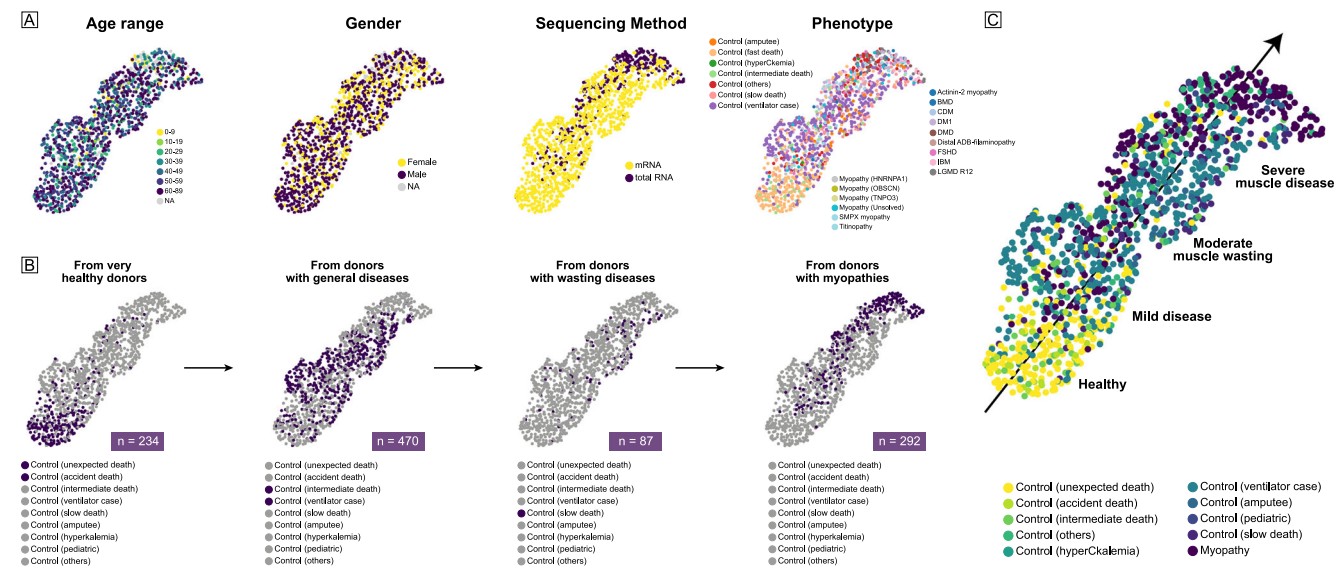

**Fig. 2 | The myopathy spectrum.** This represents a Uniform Manifold Approximation and Projection (UMAP) dimensional reduction of the integrated dataset, where muscles with similar expression patterns are closely located. **A** Both gender and age are evenly distributed across the spectrum. Myopathy muscles sequenced using different methods, including those from the GEO database (100% sequenced by total RNA) and Helsinki (100% sequenced by mRNA), are grouped together in the upper-right corner of the spectrum. **B** This spectrum order was inferred from detailed phenotypes of GTEx donors: (1) 234 very healthy donors (fast death of

natural causes or sudden unexpected deaths, e.g., car accident or suicide); (2) 470 donors with general diseases (ill but death was unexpected or ventilator using cases); (3) 87 donors with wasting diseases (slow death after a long illness, e.g., cancer or chronic pulmonary disease). Muscles located from the lower-left to the upper-right represent a range from very healthy donors to donors with general diseases, to donors with wasting diseases, and finally to myopathy donors. **C** The myopathy spectrum progresses from Healthy→Mild disease→Moderate muscle wasting→Severe muscle disease.

## Table 1 | Demographic information of the integration dataset

| Phenotype | Sample size | Sequencing method | Data source | Sex (male proportion) | Age range | | | | | | |
|---|---|---|---|---|---|---|---|---|---|---|---|
| | | | | | 0−9 | 10−19 | 20−29 | 30−39 | 40−49 | 50−59 | 60−89 |
| Control (accident death) | 31 | mRNA | GTEx | 71.0% | 0.0% | 0.0% | 25.8% | 16.1% | 16.1% | 25.8% | 16.2% |
| Control (unexpected death) | 203 | mRNA | GTEx | 79.3% | 0.0% | 0.0% | 0.5% | 2.0% | 8.9% | 38.4% | 50.2% |
| Control (intermediate death) | 46 | mRNA | GTEx | 65.2% | 0.0% | 0.0% | 2.2% | 0.0% | 6.5% | 21.7% | 69.5% |
| Control (ventilator case) | 424 | mRNA | GTEx | 63.0% | 0.0% | 0.0% | 13.4% | 12% | 20.8% | 32.3% | 21.5% |
| Control (slow death) | 87 | mRNA | GTEx | 63.2% | 0.0% | 0.0% | 0.0% | 4.6% | 9.2% | 20.7% | 65.5% |
| Control (others) | 111 | mRNA/total RNA | GTEx/GEO | 55.0% | 42.5% | 10% | 12.5% | 5.0% | 5.0% | 12.5% | 12.5% |
| Control (amputee) | 24 | mRNA | Helsinki | 64.7% | 0.0% | 0.0% | 0.0% | 0.0% | 0.0% | 0.0% | 100.0% |
| Control (hyperCkemia) | 3 | mRNA | Helsinki | NA | NA | NA | NA | NA | NA | NA | NA |
| FSHD | 61 | total RNA | GEO | 64.7% | 0.0% | 0.0% | 5.9% | 5.9% | 14.7% | 35.3% | 38.2% |
| DM1 | 44 | total RNA | GEO | 47.7% | 0.0% | 0.0% | 13.6% | 27.3% | 38.6% | 20.5% | 0.0% |
| LGMD R12 | 41 | total RNA | GEO | 100.0% | 0.0% | 0.0% | 7.3% | 29.3% | 34.1% | 12.2% | 17.1% |
| CDM | 36 | total RNA | GEO | 52.8% | 69.5% | 30.5% | 0.0% | 0.0% | 0.0% | 0.0% | 0.0% |
| Titinopathy | 31 | mRNA | Helsinki | 76.7% | 3.3% | 10.0% | 0.0% | 13.3% | 3.3% | 33.3% | 36.8% |
| IBM | 28 | mRNA/total RNA | Helsinki (GEO) | 53.6% | 0.0% | 0.0% | 0.0% | 0.0% | 0.0% | 3.6% | 96.4% |
| DMD | 5 | total RNA | GEO | NA | NA | NA | NA | NA | NA | NA | NA |
| BMD | 5 | total RNA | GEO | NA | NA | NA | NA | NA | NA | NA | NA |
| Actinin-2 myopathy | 5 | mRNA | Helsinki | 80.0% | 0.0% | 0% | 0.0% | 40.0% | 20.0% | 40.0% | 0.0% |
| Myopathy (HNRNPA1) | 5 | mRNA | Helsinki | 60.0% | 0.0% | 0.0% | 0.0% | 0.0% | 40.0% | 40.0% | 10.0% |
| SMPX myopathy | 4 | mRNA | Helsinki | 100.0% | 0.0% | 0.0% | 0.0% | 0.0% | 25.0% | 75.0% | 0.0% |
| Myopathy (OBSCN) | 1 | mRNA | Helsinki | 0.0% | 0.0% | 100.0% | 0.0% | 0.0% | 0.0% | 0.0% | 0.0% |
| Myopathy (TNPO3) | 1 | mRNA | Helsinki | 0.0% | 0.0% | 100.0% | 0.0% | 0.0% | 0.0% | 0.0% | 0.0% |
| Distal ADB-filaminopathy | 1 | mRNA | Helsinki | 100.0% | 0.0% | 0.0% | 0.0% | 0.0% | 0.0% | 0.0% | 100.0% |
| Myopathy (Unsolved) | 24 | mRNA | Helsinki | 68.3% | 17.6% | 0.0% | 0.0% | 41.2% | 0.0% | 17.6% | 23.6% |

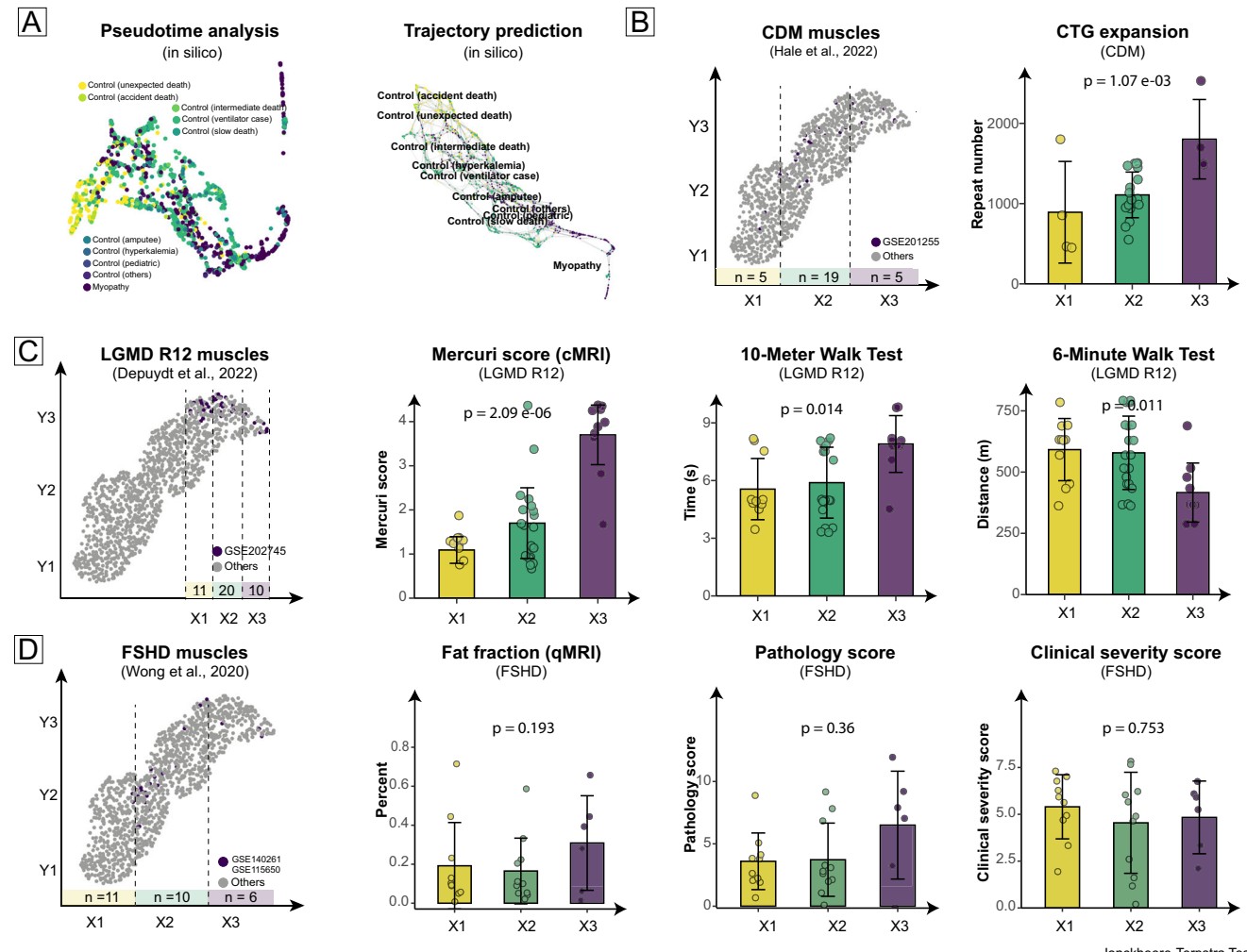

**Fig. 3 | Clinical features and transcriptional spectrum.** The spectrum order was validated using both in-silico algorithms and clinical features from different myopathies. Given that there are no linear relationships in the muscle UMAP plot, for each myopathy, we categorized the muscles into three categories by their UMAP X location and tested whether an ordered relationship exists among these three groups using the Jonckheere-Terpstra test. **A** The results of the in-silico pseudo-time and trajectory algorithms were consistent with the GTEx inference. **B** CTG expansions (in peripheral blood) increased with the spectrum progression in congenital myotonic dystrophy (CDM) patients. **C** Mercuri scores, as well as 10-meter and 6-minute walk test results, were also consistent with disease progression in limb girdle muscular dystrophies R12 (LGMD R12) patients. The more compact distribution seen in LGMD R12, as compared to CDM and FSHD, might be due to the repeated biopsies taken. For each LGMD R12 patient, three biopsies were collected (represented by three purple dots), whereas for each CDM and FSHD patient, only one biopsy was taken (represented by a single purple dot). **D** Fat fractions, pathology, and clinical severity scores of facioscapulohumeral muscular dystrophy (FSHD) patients did not show as clear a trend with disease progression as seen in CDM and LGMD R12. This is presumed to be related to the heterogeneous muscle locations (thigh and leg) in the FSHD study.

ordered relationship exists among these three groups using the Jonckheere-Terpstra test.

### Tissue deconvolution and differentially expressed analysis
A deep-learning-based autoencoder method (TAPE) was utilized to predict the cell-type composition in the integrated dataset[25]. For tissue specificity, two different skeletal muscle single-cell datasets were used as reference data in the TAPE deconvolution and were analyzed separately[26,27]. Differentially expressed gene analysis (DEG) was conducted in the classic EdgeR pipeline using the batch-adjusted count data[28]. The significant genes were selected based on both log2Foldchange ($|logFC| > 0.5$) and adjusted $p$ value (FDR < 0.05). These significant genes were further enriched in different databases (Human Phenotype Ontology, CellMarker Augmented, Kyoto Encyclopedia of Genes and Genomes, Gene Ontology, Reactome, Wiki-Pathway) for pathway analysis, which was performed using gseapy (Python).

### RNA isolation and real-time polymerase chain reaction (RT-qPCR)
Six high-ranked featured genes (MGST1, AOX1, FASN, PRKCD for general myopathy, CD163 for IBM, and CYP4B1 for titinopathy) were validated in our local muscles, and primer information can be found in Supplementary Table 1. In further detail, RNA was extracted from lower leg muscle biopsies of 13 patients and 6 controls using the Qiagen RNeasy Plus Universal Mini Kit (Qiagen, Hilden, Germany) according to the manufacturer's instructions. cDNA synthesis was performed using SuperScript III Reverse Transcriptase (Invitrogen TM) and random primers, according to the manufacturer's protocol. RT-qPCR assays were performed using the iQ SYBR Green Supermix (BIO-RAD) and 25 nM of each specific primer. Each assay was performed with technical triplicates for each of the biological samples. To normalize, 18S was used as the reference gene. The results were calculated in double Delta Ct method and presented in relative quantification (RQ) form.

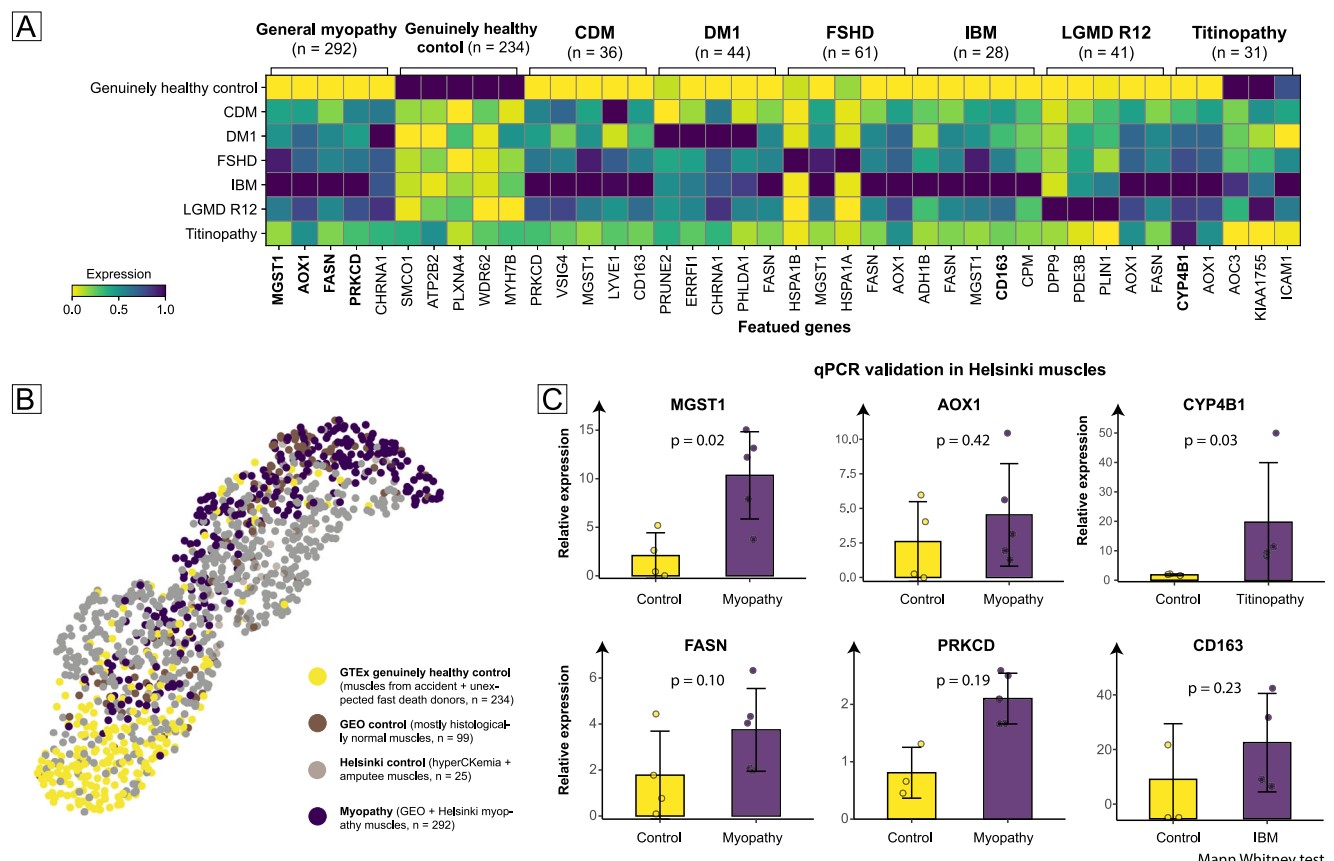

**Fig. 4 | Featured genes and qPCR validation. A** The top five up-regulated genes in general myopathy and six myopathies with more than 25 muscles are shown. These are compared sequentially with the genuinely healthy control group ($n = 234$). **B** Since both the GEO and Helsinki controls, which were used in previous myopathy studies, were located in the "Wasting" state of the myopathy spectrum, we chose to use genuinely healthy muscles from the GTEx as unprecedented controls. This provided an alternative perspective on myopathy and informed the creation of figure (**A**). **C** Featured genes in general myopathy (MGST1, AOX1, FASN, PRKCD), titinopathy (CYP4B1), and IBM (CD163) were validated in Helsinki myopathy and control muscles (randomly selected). The results were compared using the Mann−Whitney test.

## Statistics and reproducibility

Python (version 3.8.1) and R (version 4.2.2) were used to analyze the data. All relevant code and data used in this study have been deposited on GitHub (https://github.com/Hirriririir/Myopathy-Spectrum) and Zenodo[29].

## Reporting summary

Further information on research design is available in the Nature Portfolio Reporting Summary linked to this article.

## Results

### A myopathy spectrum revealed after integration

The integrated datasets included muscles from 292 individuals with myopathies (Fig. 2): facioscapulohumeral muscular dystrophy (FSHD, $n = 61$), myotonic dystrophy type 1 (DM1, $n = 44$), limb girdle muscular dystrophies R12 (LGMD R12, $n = 41$), congenital myotonic dystrophy (CDM, $n = 36$), titinopathy ($n = 31$), inclusion body myositis (IBM, $n = 28$), Duchenne muscular dystrophy (DMD, $n = 5$), Becker muscular dystrophy (BMD, $n = 5$), actinin-2 myopathy ($n = 5$), HNRNPA1 myopathy ($n = 5$), SMPX myopathy ($n = 4$), OBSCN myopathy ($n = 1$), TNPO3-myopathy ($n = 1$), distal ADB-filaminopathy ($n = 1$), and unsolved myopathy ($n = 24$) (Table 1). The control groups included muscles from 929 controls: accident death control ($n = 31$), unexpected death control ($n = 203$), intermediate death control ($n = 46$), ventilator case control ($n = 424$), slow death control ($n = 87$), other controls ($n = 111$), amputee control ($n = 24$), and hyperCkemia control ($n = 3$). The female to male ratio is 363:737, and the sex information is not available for 121 donors. The age range is as follows: 0−9 years ($n = 46$), 10−19 years ($n = 20$), 20−29 years ($n = 83$), 30−39 years ($n = 105$), 40−49 years ($n = 165$), 50−59 years ($n = 303$), 60−89 years ($n = 389$). Age information is not available for 119 donors.

After integration, all muscles from different sources and sequenced by different methods were harmonized together in the PCA analysis by adjusted read counts, clearly contrasting with the previous state by original read counts (Supplementary Fig. 1). It is noteworthy that most myopathy muscles from both GEO (100% sequenced by total RNA) and Helsinki (100% sequenced by mRNA) were located at the right corner of the UMAP plot. All of this indicates that the batch-effect among different studies may have been well-adjusted. Considering the heterogeneous health status within the control muscles used in the myopathy studies (GEO and Helsinki), we re-classified the GTEx muscle donors into three control groups: (1) 234 very healthy donors (fast death of natural causes or sudden unexpected deaths, e.g., car accident or suicide); (2) 470 donors with mild diseases (ill but death was unexpected or ventilator using cases); (3) 87 donors with moderate muscle wasting (slow death after a long illness, e.g., cancer or chronic pulmonary disease). With the detailed classification of control muscles, a potential myopathy spectrum order was reflected: Healthy→Mild disease→Moderate muscle wasting →Severe muscle disease. Interestingly, unlike single-cell analysis where similar cell types cluster together, the distribution of myopathy muscles in our study is not as compact. We observed a ribbon-like intrusion of myopathy muscles into the healthy and general disease group. This pattern is analogous to the asymptomatic preclinical stages observed in clinical practice.

## Spectrum order validation with clinical features

This spectrum order was first validated by in-silico analyses (Fig. 3A). Both pseudo-time analysis and trajectory prediction algorithms provide similar muscle deterioration transformation to the severity spectrum. Interestingly, 21 pediatric control muscles, which were specifically isolated from the "Control (others)" category, were closely located to the slow death controls in the trajectory prediction result. This is a finding quite contrary to common expectations (muscles from control children should be healthier). We examined the original article and discovered that these were histopathologically normal muscles, suggesting their health status might be comparable to most GEO controls and our amputee and hyperCkemia controls[17]. Then, clinical features of three myopathies (CDM, LGMD R12, and FSHD) were analyzed[14,17,18]. Repeated muscle biopsies taken from donors can disturb the distribution. For instance, three biopsies were taken from three muscles in each LGMD R12 donor, hence the distribution of LGMD R12 was a compact oval, contrasted with the elongated ribbons in CDM and FSHD (where only one biopsy was taken from each patient).

The clinical features of myopathy donors also generally correlated with the severity spectrum order (Healthy→Mild disease→Moderate muscle wasting→Severe muscle disease). CTG repeat sizes (peripheral blood) showed increased expansion in the spectrum order ($n = 29$): JT = 181, $p = 1.07$ e-03 (Jonckheere-Terpstra test) in Fig. 3B. In LGMD R12, Mercuri score (a semi-quantitative scoring system for muscular fat infiltration, JT = 459, $p = 2.09$e-06), 10-meter walk test (JT = 369, $p = 0.011$), and 6-min walk test (JT = 164, $p = 0.014$) also exhibited statistically significant increased order in these three groups (Fig. 3C). However, this increased order was not evident in FSHD muscles: fat fraction (a quantitative calculation of muscular fat infiltration using specific MRI series, JT = 139, $p = 0.36$), pathology score (JT = 147, $p = 0.19$), and clinical severity score (JT = 125, $p = 0.75$) (Fig. 3D).

## Tissue deconvolution showed shared features among myopathies

Two different skeletal muscle single-cell datasets were utilized to decipher the tissue deconvolution in the integrated dataset. The cell composition of five controls (accident death ($n = 31$), unexpected death ($n = 203$), intermediate death ($n = 46$), ventilator case ($n = 424$), slow death ($n = 87$)) was compared with the myopathy groups. Although these were in-silico prediction results, the first Tabula Sapiens dataset (30,746 cells) found fewer vasculature structure cell proportions (vascular tree endothelial cells and pericytes), and a higher proportion of satellite stem cells, natural killer (NK) T cells, and fibroblast (tendon) cells in the myopathy groups (Supplementary Fig. 2). Similarly, using another skeletal muscle single-cell dataset GSE143704 ($n = 22,058$) as a reference, fewer vasculature structure cell proportions (endothelial cells and pericytes), and a greater proportion of adipocytes and COL1A+ fibroblasts were found in the myopathy groups (Supplementary Fig. 3). Note that the proportion deconvoluted can be significantly impacted by biopsy selection; therefore, minor differences should be interpreted with caution.

## Featured transcriptional genes and pathways using unprecedented control healthy muscles

After integration, an obvious phenomenon observed in the UMAP plot was that the control muscles used in almost all myopathy studies (GEO and Helsinki) were predominantly located within the moderate and severe muscle wasting stages when compared with the GTEx muscles as a reference (Fig. 4B). This is consistent with clinical practice; since doctors most often cannot obtain healthy muscles due to ethical reasons and instead use histologically normal muscles to represent 'control' muscles. Considering these 'control' muscles are not genuinely healthy and may introduce some biases, we utilized the GTEx's genuinely healthy muscles (from accident and unexpected death cases, $n = 234$) as controls to provide a new perspective on these myopathies (Supplementary Data 2).

General myopathy ($n = 292$) and six different myopathies (each with a sample size >25) were selected for the DEG analysis (Fig. 4A). The six different myopathies included CDM ($n = 36$), DM1 ($n = 44$), FSHD ($n = 61$), IBM ($n = 28$), LGMD R12 ($n = 41$), and titinopathy ($n = 31$). For all myopathies, 200 up-regulated genes and 568 down-regulated genes were revealed. The top five up-regulated genes were *MGST1*, *AOX1*, *FASN*, *PRKCD*, and *CHRNA1* (logFC > 1.28, FDR < 1.24 e-22), while the top-ranked down-regulated genes were *SMOC1*, *ATP2B2*, *PLXNA4*, *WDR62*, and *MYH7B* (logFC < −1.09, FDR < 4.09 e-29). Pathway analysis of the DEGs underscored changes in the myopathy muscles, including biophysiological pathways like muscle contraction, lipoatrophy, myotube cell involvement, and FATZ (filamin-, α-actinin-, and telethonin-binding protein of the Z-disk) binding (Supplementary Fig. 4). These pathway annotations are of interest and will be further analyzed (Supplementary Figs. 5−10).

For validating the consistency of the integration results to previous studies, we reanalyzed the original and batch-adjusted read counts using the same processing pipeline (EdgeR) and the same criteria of |logFC| > 0.5 and FDR < 0.05. We then compared the differences between the two sets of results. The overlap of DEG genes between the integrated dataset and the original studies varied from 4.2% to 18.1% across the FSHD, LGMD R12, IBM, and CDM groups (Supplementary Fig. 11). However, the myopathy-featured genes (top 15 upregulated) identified by the integration dataset (based on genuinely healthy muscles from GTEx) were consistently preserved in each myopathy and the $p$ values drastically decreased with enlarged sample sizes, even when compared with different control groups. Specifically, 80% of the featured genes were preserved in FSHD (adult control, Supplementary Table 2), 80% in LGMD R12 (adult control, Supplementary Table 3), 73.3% in IBM (amputee control, Supplementary Table 4), and 13.3% in CDM (pediatric control, Supplementary Table 5). Interestingly, when we enlarged our previous IBM data (GSE151757, IBM vs. amputee control = 24:9) using updated samples from our center (IBM vs. amputee control = 28:24), we found that the preservation of the top featured genes increased to 100%. This was a similar case in the re-analysis of the original data from the longitudinal FSHD follow-up studies conducted by Wang et al.

## qPCR validation in Helsinki muscles

The batch correction step in data processing may influence the final DEG results; hence, we used the qPCR method to further validate these newly identified high-ranked genes (Fig. 4C). We selected four up-regulated genes for general myopathy (*MGST1*, *AOX1*, *FASN*, *PRKCD*), *CD163* for IBM, and *CYP4B1* for titinopathy, aiming to validate their actual expression in our local muscles. Since genuinely healthy muscles were also unavailable to us due to ethical reasons, we randomly selected four amputee and two hyperCkemia muscles as controls[30]. Comparatively, five general myopathy muscles (two Actinin-2 myopathy and three IBM), four IBM muscles, and four titinopathy muscles were selected as experimental groups (Supplementary Data 3). The genes featured for general myopathy indeed showed relatively higher expression trend in the myopathy muscles: *MGST1* (RQ 10.34/2.09, $p = 0.02$), *AOX1* (RQ 4.53/2.41, $p = 0.42$), *FASN* (RQ 3.75/1.75, $p = 0.10$), *PRKCD* (RQ 2.10/1.23, $p = 0.19$). Similarly, titinopathy featured *CYP4B1* showed higher expression in titinopathy muscles (RQ 19.71/1.41, $p = 0.03$), and IBM featured *CD163* showed higher expression in IBM muscles (RQ 27.87/10.05, $p = 0.23$).

## Discussion

This integrative study provides a general view of the "Healthy→Mild disease→Moderate muscle wasting→Severe muscle disease" transformation in muscle diseases. Firstly, we bridged the gap between muscle bulk-RNA sequencing and the phenotypes associated with muscle deterioration. Secondly, by utilizing the GTEx genuinely healthy muscles as an unprecedented control, we identified new featured genes in myopathies, which offer a novel

perspective on the usage of muscle bulk-RNA sequencing in muscle diseases.

This transcriptional spectrum correlates well with actual clinical progression in muscle diseases. Given the randomness of sex and age as shown in Fig. 2, it is unlikely that these demographic factors are the cause of the observed spectrum order. Due to the hereditary nature of most myopathies, they usually begin insidiously during life, if not present at birth. Their clinical spectrum is versatile, starting from changes in the muscle itself (genetic, myofiber type, pathology) to the overall capabilities of a patient (force, endurance, involvement of other organs)[31–33]. In this study, the muscles from CDM and LGMD R12 patients showed a strong correlation with clinical features, which was not statistically evident in FSHD muscles. We suspect this may be related to the muscle selection for the RNA-seq and the large variability of different and often asymmetric muscle involvements between patients. FSHD was also more heterogeneous in terms of muscle selection: one biopsy was taken from each patient, either from the tibialis anterior, gastrocnemius muscles, vastus lateralis, or biceps femoris muscle[14]. The muscles were more homogeneous in the CDM (where one vastus lateralis biopsy was taken from each patient) and LGMD R12 (where three biopsies of semimembranosus, vastus lateralis, and rectus femoris muscles were taken from each patient) studies[17,18]. However, muscle physiology may not correlate with overall clinical manifestations, because biopsy selection is not a standardized process, and muscle involvement has its own specific patterns that vary among different myopathies[34].

The selection of controls can greatly impact the muscle RNA-seq results. Since genuinely healthy muscle samples are difficult to obtain due to ethical reasons, we propose that these 234 GTEx muscle samples from accident and unexpected death donors could provide a reasonable source[12]. Our study not only supports the results regarding clinical progression but also complements these excellent original work on CDM, DM1, FSHD, LGMD R12, IBM, as new featured genes were identified using our method[13,14,16–19]. Typically, the top 10−20 ranked genes are considered the most important in an RNA-seq dataset, analogous to a pyramid structure. The featured genes identified by our integration dataset consistently persist when compared with different control phenotypes, which may indicate that the tip of the pyramid remains relatively constant (with larger overlapping in the featured genes), while the bottom can be easily disrupted (smaller overlapping in all DEG genes). The relatively smaller overlapping of featured genes in the case of CDM can be explained by the age difference in the control samples. The control samples used in the original CDM study consist of histologically normal pediatric muscles (0−9 years old), whereas the GTEx genuine healthy muscles are all >20 years old[17].

Our integrated dataset has helped identify common and specific featured genes and pathways for myopathies. The top five up-regulated genes in general myopathy are MGST1, AOX1, FASN, PRKCD, and CHRNA1, which have been rarely reported in previous myopathy studies. MGST1 (Microsomal Glutathione S-Transferase 1) has been reported for its potential role in protecting against oxidative stress and in aging[35]. AOX1 (Aldehyde Oxidases 1) has been identified as a contributor to myogenesis[36]. The role of FASN (fatty acid synthase) is evident, as muscular fat infiltration is common in myopathic muscles. PRKCD (Protein Kinase C delta) also participates in the regulation of lipogenesis[37]. The functions of CHRNA1 (Cholinergic Receptor Nicotinic Alpha 1 Subunit) have been thoroughly investigated in the elderly and aging rodents, as its levels increase in aging skeletal muscle. Upregulation of CHRNA1 can also induce and aggravate sarcopenia[38]. The elevated expression of CYP4B1 may be associated with altered fatty acid metabolism, while increased levels of CD163 could correspond to an enrichment of CD163+ macrophages in IBM muscles[39,40]. Additionally, CD163 has shown increased expression with statistical significance in our previous work and in the study conducted by Hamann et al.[19,41]. Pathway enrichment analysis also confirms the clinical understanding of

myopathy, such as muscle contraction impairment, and FATZ binding (FATZ forms a tight complex and phase-separated condensates with α-actinin)[42]. However, some detailed pathomechanisms in these myopathies are not easily categorized with the currently available knowledge.

## Limitation
This study has several limitations. First, numerous biases may be introduced during data integration, including those related to biopsy site and muscle selection, control selection, bulk RNA-seq method selection, and sample size. Additionally, the batch adjustment algorithm has altered the count data from the original studies. Second, sample selection and cell type classifications as defined in the original single-cell dataset may also lead to misinterpretations of the predicted deconvolution results. Therefore, flow cytometry or large-scale single-cell analysis are recommended to address this issue more effectively. Third, genuinely healthy control muscles are currently unavailable for qPCR validation. Control muscles with healthier physiological states should be considered in future studies. Fourth, due to biopsy sample restrictions, we could only validate the featured gene expression in titinopathy and IBM. More genes and myopathies can be validated in multi-center studies. Fifth, clinical information is not available for most muscles in the integrated dataset. Encouraging data sharing and collaboration could facilitate future myopathy studies.

## Conclusions
In summary, we created an integrated human skeletal muscle bulk-RNA-seq dataset (1221 muscles × 9231 genes) by combining public datasets with our local data. The myopathy spectrum (Healthy→Mild disease→Moderate muscle wasting→Severe muscle disease) was revealed upon integration, and the clinical features of myopathies were well-correlated with the spectrum order. By utilizing genuinely healthy muscles as unprecedented control samples, we provided an alternative perspective for deciphering changes in the studied myopathies. This approach identified new featured genes and pathways not only generally in myopathies but also for specific types of myopathies.

## Data availability
The source data for graphs and figures is available on GitHub (https://github.com/Hirririiriir/Myopathy-Spectrum). The original data of each public datasets used in this study are available in their original research (links are also provided in the Myopathy-Spectrum GitHub repository).

## Code availability
The analyzing code for the integration dataset is available on GitHub (https://github.com/Hirririiriir/Myopathy-Spectrum).

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

## Acknowledgements

We thank the participants and their families who donated their muscle tissues for research purposes. We would also like to extend special thanks to the authors of these publicly available muscle datasets, which will facilitate further research in the future. Funding support: FHRC (M.S. and M.J.), Academy of Finland (M.S.), Jane and Aatos Erkko foundation (P.H.), Magnus Ehrnrooth foundation (A.O.), AFM Téléthon (M.J.), China Scholarship Council (H.Z.). The funders had no role in the design and conduct of the study; collection, management, analysis, and interpretation of the data; preparation, review, or approval of the manuscript; and decision to submit the manuscript for publication.

## Author contributions

H.Z., M.S., M.J., A.O. and P.H.J conceived and designed the study. H.Z. and M.J. performed data analyses. V.S. performed validation experiment. S.K.

and A.O. provided statistical support. H.Z. drafted the original manuscript. B.U. and P.H. critically revised the draft manuscript. All authors approved the final version of the manuscript.

## Competing interests

The authors declare no competing interests.
