## [Peer Review File · Communications Biology]

Reviewers' comments:

Reviewer #1 (Remarks to the Author):

The study by Zhong et al intends to establish a molecular myopathy spectrum correlated to clinical severity. Indeed the number of samples included in this study, either in silico or "live", is very important and strongly supports the reliability of the results. This study is a very impressive bioinformatic exercise, that gives us a comprehensive view of the myopathy kinetics, but it does not unravel really new pathways for myopathy, such as oxidative stress, myogenesis, muscle contraction impairment, fatty acid metabolism, etc... that have been already reported in many studies on muscle diseases. I also do not see practical implications for either diagnosis or treatment, except for the targeting of some of these most likely secondary effects, which is already been under investigation in some studies. In summary, the implications of this established spectrum should be strongly emphasized to show its utility. It could be that I have totally missed the message, meaning that it should be much better explained.

Reviewer #2 (Remarks to the Author):

In the manuscript entitled "Revealing Myopathy Spectrum: Integrating Transcriptional and Clinical Features of Human Skeletal Muscles with Varying Health Conditions", Zhong et al integrate RNA sequencing data of hundreds of skeletal muscle tissues from multiple sources, including healthy controls and different myopathies. They integrate the data in order to account for factors such as methodology and cohort. The obtained UMAP allowed the authors to identify cluster of genuine healthy controls and myopathic patients and a spectrum of early non-myopathic cases or not genuine healthy controls. They also show how the gene expression signature related to clinical and MRI readouts. They further identify a common set of dysregulated genes in myopathic cases and use qPCR to try to validate such gene set. They also mention limitation of the study such as the presence of different muscle groups, sequencing method and sample size.

The work is in my opinion very interesting and needed in the field. Integration of large number of samples provides sufficient power to identify myopathic features and allows a better view of the processes occurring in disease.

I would have a few points that in my opinion need clarification and I hope they will make the manuscript stronger.

Major points

1. Data availability for future research.

a. The collection and integration of such data is a time consuming and hardly needed exercise. The analyses proposed cover part of the analyses that could be performed with this dataset. Other researchers might want to use such integrated dataset to allow comparison with in house data. I think the impact of this work will be much larger if the integrated would be made available online, together with the possibility to upload own data.

b. Integrated data in github seem to still be count (integer) data. Could the authors explain why this is not normalized?

c.

2. Figure 3

a. Panel A Trajectory / pseudotime – such techniques are have based on spliced and unspliced reads? Are the results affected by the fact some samples are obtained from mRNA seq instead of total RNA? What are the genes mostly contributing to the pseudotime? Are there for example common transcription factors explaining pseudotime or converting healthy to myopathy? Is there any overlap

with pseudotime / dux4 genes in fshd? Or with cd8 in IBM?

b. Grouping in panel B-D. Separation in 3 groups seems to be based on X axis? Myopathic phenotype seems to be captured by the Y axis too. Why were vertical axes chosen for grouping? Number cases seems to be a subset of the total number of cases available. Analysis is performed in n=29 CDM, while 36 are reported in table 1. For LGMD this is correct (n=41), while for FSHD there is also a discrepancy (n=27 in figure 3 and n=61 in table 1). What is the reason for different n? This should be better explained in the manuscript.

c. Lines 194-195: The clinical features of myopathy donors also generally correlated with the severity spectrum. What is meant by clinical features and severity spectrum order? Different myopathies will present with different clinical features. Where is this presented?

d. Line 195-196 Blood samples analysis does not appear in the methods. This should be added.

e. Repeated measurements in LGMD R12: was the within subject variation modelled? Or where samples considered as independent? Are samples obtained from same cases closer than across individuals? This cannot be appreciated in the figure. It is not clear what is meant by oval distribution at line 191. Does this refer to figure 3c?

f. How was the mapping of clinical and MRI scores to the UMAP done? I understand such data were obtained from supplementary data of original studies. Was it possible to match the data to the individual? Does this perhaps explain the reduction in n compared to the table? If so, I would clarify this and include the clinical data into the integrated data or provide an extra table with such pairing.

g. Would it be possible to provide more insight on how the myopathic signature relates to clinical signature. What genes are contributing the most to the UMAP?

3. Deconvolution. eFigs 2 and 3.

a. Line 207: If I understand well 5 controls per group were chosen for deconvolution. Why choosing a selection and not all? And how were the 5 controls cases chosen for each group? Could you please clarify here and in the manuscript?

b. I believe proportions of cells are plotted and they are expressed in total %. I would suggest to not underline minor differences of a few % points as these could be more related to sampling rather than biological differences. For example the presence of a blood vessel could skew such proportions.

c. It is unclear how the comparison was done: myopathy vs rest (healthy controls pooled)? If so, why?

d. I also wonder whether certain cell types (like CD8) were found to be more present in IBM samples where the inflammatory signature and presence of CD8 cells is expected.

e. Certain relevant cell types such as fibroadipogenic progenitor cells seem to be totally absent? What is the reason for that?

f. To what extent the lack of disease specific single cell dataset affects the deconvolution? Perhaps this comparison could be made for 1 disease for which single cell data are available. I also wonder whether the proportion of cells in the single cell dataset affects the proportions estimate by TAPE. I would suggest to evaluate this changing the proportions of cells in the single cell datasets by down-sampling cell populations.

4. Figure 2. Line 166 does not seem only right corner, there seems to be a stripe going down showing overlap with donors with general disease. Perhaps related to some non-specific inflammation? or spectrum asymptomatic pre-clinical stages as mentioned at line 177?

5. Figure 4

a. Fig4a why the numbers of each group are given on top of the figure? Columns identify genes. Shouldn't the n be mentioned on the left for each row?

b. Fig4b color use – middle 2 groups difficult to distinguish

6. Contribution of different muscle types.

a. One of the limitation is the muscle selection. There are 2 studies published where multiple paired muscles were analyzed and sequenced. 10.3389/fgene.2023.1216066 and 10.7554/eLife.80500. Would it be possible to use those datasets to estimate the effect of muscle group and or

oxidative/glycolytic muscle?

b. Does the data allow to identify genes associated with the patterns of muscles involved in different myopathies?

7. qPCR: how was the statistics done?

8. Discussion

a. The authors mention in the abstracts (lines 34-36) and in the discussion that they bridged the gap between muscle bulk-RNA sequencing and clinical features. I find this to be not represented in the data. The authors show that patients cluster on specific areas based on their gene expression and that relates to some extent in LGMD R12 patients (mostly for the X3 group), however this was not the case for other diseases. A clear linear relationships between gene expression and clinical features was not reported. I therefore suggest amend such sentences.

b. I would also suggest to discuss why healthy pediatric cases clustered with some pathological samples

Minor points

9. Line 58 add reference

10. Line 173-177 the sentence is not clear. Could this be clarified?

11. Fig 1 CTG expansion – expansion

12. Supplement 3 (line 227) seems to be missing

Reviewer #3 (Remarks to the Author):

Xiao et al. present a very interesting work transcriptional and clinical spectrum of a selection of myopathies. Even though the manuscript is definitely of interest to the field, some pieces of information are missing.

Line 36: CTG expansion should be at least once clarified with an appropriate gene name.

Line 87 and 147-160: The myopathy diagnoses should be provided in the Materials and Methods section rather than in Results, as the diagnoses were not the result of the presented work. The particular muscles taken for the biopsy should also be presented, at least for control and myopathy subgroups.

Line 103-104: The gene numbers should be precisely described. I presume 9231 genes had a coverage of at least one in every sample. How were 16953 candidate genes selected? I presume these are either muscle-specific genes or had coverage of at least one in at least one sample. If these are muscle-specific genes, why the expression of almost half of them was undetectable in every sample?

Line 111, or rather 180-182: The authors should explain what was the reason behind the pseudo-time analysis and trajectory prediction. Did you want to check if control muscles are in the process of aging/degeneration/wasting due to age, any other disease, or additional factors?

Line 124: How many samples were in the single-cell datasets?

Line 157: HyperCKemia does not seem an appropriate control as it is usually a sign of muscle damage.

Line 173-175: The mild disease and moderate muscle wasting groups should be precisely described. Are these subgroups of myopathy group with lower UMAP scores of clinical features? If not, how were they performing in Pseudotime analysis and trajectory prediction? Alternatively, are these subgroups of healthy donors, but those with general diseases and wasting diseases respectively? Could all subgroups, from both healthy and myopathic muscle RNA-seq, be presented in such a spectrum? In addition, was there any correlation between pseudo-time analysis or trajectory prediction and the age of the donors?

Line 234-235: The differently-expressed genes on which pathway analysis was performed should be briefly described (an arbitrarily selected number of top-ranked genes? or rather genes below the given FDR threshold) in the manuscript and listed in the supplementary material.

Line 560: The subtotal of the demographic information (male proportion and age range) should be provided for the controls (8 control groups together) and myopathies (15 myopathy groups).

Response to reviewer #1

1. The study by Zhong et al intends to establish a molecular myopathy spectrum correlated to clinical severity. Indeed the number of samples included in this study, either in silico or "live", is very important and strongly supports the reliability of the results.

Response: Thank you for acknowledging our work.

2. This study is a very impressive bioinformatic exercise, that gives us a comprehensive view of the myopathy kinetics, but it does not unravel really new pathways for myopathy, such as oxidative stress, myogenesis, muscle contraction impairment, fatty acid metabolism, etc... that have been already reported in many studies on muscle diseases.

Response: Thank you for your comments. We acknowledge the key innovation points of our work:

1. The integration and analysis of a large dataset of human skeletal muscles is a novel approach in rare disease research.
2. While many pathways have been previously reported, it is interesting that most of the top-ranked genes identified by this study have not been reported in prior myopathy studies, although some were noted in muscle physiology research. These potentially overlooked genes could play significant roles in the pathogenesis of various myopathies.
3. Our study offers valuable insights for future transcriptional research in muscle diseases. We emphasize the importance of control selection and have identified a novel group of 'very healthy' control muscles (consisting of GTEx muscles from individuals who died in accidents or suddenly, $n = 234$), which could serve as a robust reference for future studies.
4. Additionally, all data from our study is publicly available. We believe this dataset will facilitate future research into muscle diseases.

3. I also do not see practical implications for either diagnosis or treatment, except for the targeting of some of these most likely secondary effects, which is already been under investigation in some studies.

Response: We think this is mostly related to the gene selection methodology in this study. We selected 9,231 muscle-specific genes based on their presence (with a count of at least one) in every sample ($n = 1221$), which resulted in focusing on genes highly expressed in both myopathy and control samples. Consequently, some genes that are indicative of detrimental effects, such as DUX4 in FSHD, were not included.

The current work represented preliminary data analysis, primarily focusing on the overall state of all muscles. We try to prove that these large group of transcriptome datasets can be integrated first. Our future studies will delve more deeply into the myopathy muscles.

4. In summary, the implications of this established spectrum should be strongly emphasized to show its utility. It could be that I have totally missed the message, meaning that it should be much better explained.

Response: Thank you for your suggestion. We did some modification to the reverent parts:

The overlap of DEG genes between the integrated dataset and the original studies varied from 4.2% to 18.1% across the FSHD, LGMD R12, IBM, and CDM groups (eFigure 11 in Supplement 2). However, the myopathy-featured genes (top 15 upregulated) identified by the integration dataset (based on genuinely healthy muscles from GTEx) were consistently preserved in each myopathy and the p values drastically decreased with enlarged sample sizes, even when compared with different control groups. Specifically, 80% of the featured genes were preserved in FSHD (adult control, eTable 1 in Supplement 2), 80% in LGMD R12 (adult control, eTable 2 in Supplement 2), 73.3% in IBM (amputee control, eTable 3 in Supplement 2), and 13.3% in CDM (pediatric control, eTable 4 in Supplement 2). Interestingly, when we enlarged our previous IBM data (GSE151757, IBM vs. amputee control = 24:9) using updated samples from our center (IBM vs. amputee control = 28:24), we found that the preservation of the top featured genes increased to 100%. This was a similar case in the re-analysis of the original data from

the longitudinal FSHD follow-up studies conducted by Wang et al. [Results. Featured transcriptional genes and pathways using unprecedented control healthy muscles, Paragraph 3]

Our integrated dataset has helped identify common and specific featured genes and pathways for myopathies. The top five up-regulated genes in general myopathy are MGST1, AOX1, FASN, PRKCD, and CHRNA1, which have been rarely reported in previous myopathy studies. [Discussion, Paragraph 1]

By utilizing genuinely healthy muscles as unprecedented control samples, we provided an alternative perspective for deciphering changes in the studied myopathies. This approach identified new featured genes and pathways not only generally in myopathies but also for specific types of myopathies. [Conclusion]

Response to reviewer #2

In the manuscript entitled “Revealing Myopathy Spectrum: Integrating Transcriptional and Clinical Features of Human Skeletal Muscles with Varying Health Conditions”, Zhong et al integrate RNA sequencing data of hundreds of skeletal muscle tissues from multiple sources, including healthy controls and different myopathies. They integrate the data in order to account for factors such as methodology and cohort. The obtained UMAP allowed the authors to identify cluster of genuine healthy controls and myopathic patients and a spectrum of early non-myopathic cases or not genuine healthy controls. They also show how the gene expression signature related to clinical and MRI readouts. They further identify a common set of dysregulated genes in myopathic cases and use qPCR to try to validate such gene set. They also mention limitation of the study such as the presence of different muscle groups, sequencing method and sample size.

The work is in my opinion very interesting and needed in the field. Integration of large number of samples provides sufficient power to identify myopathic features and allows a better view of the processes occurring in disease.

Response: Thank you for acknowledging our work. The current work represented preliminary data analysis, primarily focusing on the overall state of all muscles. Our future studies will delve more deeply into the myopathy muscles.

I would have a few points that in my opinion need clarification and I hope they will make the manuscript stronger.

Major points

1. Data availability for future research.

a. The collection and integration of such data is a time consuming and hardly needed exercise. The analyses proposed cover part of the analyses that could be performed with this dataset. Other researchers might want to use such integrated dataset to allow comparison with in house data. I think the impact of this work will be much larger if the integrated would be made available online, together with the possibility to upload own data.

Response: We concur with your perspective. Indeed, integrating sufficient clinical and genetic data is crucial for producing more robust results in the myopathy field and for fostering future advancements. All bulk RNA-seq count data from our study, including both raw and processed forms, are accessible through the link in the 'Data Sharing Statement' section of our article, available at <https://github.com/Hirriririr/Myopathy-Spectrum>.

b. Integrated data in github seem to still be count (integer) data. Could the authors explain why this is not normalized?

Response: Thank you for your question regarding the data format. It's important to note that there are various methods for RNA-seq analysis, with limma, EdgeR, and DESeq2 being among the most popular (as detailed in <https://app.jove.com/t/62528/three-differential->

expression-analysis-methods-for-rna-sequencing-limma-edger-deseq2). While normalization is a necessary step in EdgeR and limma, it is not required for DESeq2 analysis. These packages employ different normalization algorithms, which is a significant consideration.

We have chosen to provide the integer count data because it offers flexibility, allowing the data to be analyzed using almost any preferred method. However, recognizing the need for normalized data, we have also uploaded the EdgeR-normalized data for this study. This file is available in the GitHub Release section under the name 'All_muscle_L_combatseq_tmm.csv' (<https://github.com/Hirriririir/Myopathy-Spectrum/releases/tag/1.0>).

2. Figure 3

a. Panel A Trajectory / pseudotime – such techniques are have based on spliced and unspliced reads? Are the results affected by the fact some samples are obtained from mRNA seq instead of total RNA? What are the genes mostly contributing to the pseudotime? Are there for example common transcription factors explaining pseudotime or converting healthy to myopathy? Is there any overlap with pseudotime / dux4 genes in fshd? Or with cd8 in IBM?

Response: Your questions are very interesting. However, spliced and unspliced reads (used for RNA velocity calculation) can only be derived from raw sequencing data, which is unavailable for most muscles in this study. On the other hand, trajectory and pseudotime analysis only require count data and can reflect the relative activity or progression of a biological process. In contrast to RNA velocity, trajectory and pseudotime algorithms may not perform many advanced analyses, such as prioritizing driven genes.

b. Grouping in panel B-D. Separation in 3 groups seems to be based on X axis? Myopathic phenotype seems to be captured by the Y axis too. Why were vertical axes chosen for grouping? Number cases seems to be a subset of the total number of cases available. Analysis is performed in n=29 CDM, while 36 are reported in table 1. For LGMD this is correct (n=41), while for FSHD there is also a discrepancy (n=27 in figure 3 and n=61 in table 1). What is the reason for different n? This should be better explained in the manuscript.

Response: Yes, the separation into three groups was based on the X-axis of the UMAP, while myopathic phenotypes (such as CTG expansion, Mercuri score, 10MWT, etc.) were categorized based on the clinical data provided by the original studies. We believe that either the X or Y axis of the UMAP could be used for grouping. We chose X-axis purely because it is more visually intuitive.

Some clinical information was missing for the patients with sequenced muscles. Out of the 36 sequenced CDM muscles shared in GSE201255, only 29 muscles contained CTG repeats according to their original study (10.1093/hmg/ddac254, supplemental_table1). The same issue applies to the FSHD case (10.1093/hmg/ddaa031, suppl_table_1).

We have updated the caption of Fig. 3 for enhanced clarity:

Clinical features including CTG repeat number, Mercuri score from conventional MRI (cMRI), fat fraction from quantitative MRI (qMRI), pathology and inflammation scores, clinical severity score, 10-meter walk test, and 6-minute walk test were mapped onto the integral UMAP result to validate the myopathy spectrum. Note that only sequenced muscles with available clinical information were analyzed. [Methods, Mapping the UMAP spectrum with clinical features from different myopathies, Paragraph 1]

c. Lines 194-195: The clinical features of myopathy donors also generally correlated with the severity spectrum. What is meant by clinical features and severity spectrum order? Different myopathies will present with different clinical features. Where is this presented?

Response: Apology for any confusion caused here. What we want to express is that this myopathy spectrum tendency (Healthy—>Mild disease—>Moderate muscle wasting y—>Severe muscle disease) is consistent across CDM, LGMDR12, and FSHD, even when considering their distinct clinical features. While it's true that different myopathies present with their own unique features, this overarching trend of progression in severity remains generally preserved across these conditions.

We have added some modifications here:

The clinical features of myopathy donors also generally correlated with the severity spectrum order (Healthy→Mild disease→Moderate muscle wasting →Severe muscle disease). [Results, Spectrum order validation with clinical features, Paragraph 2]

d. Line 195-196 Blood samples analysis does not appear in the methods. This should be added.

Response:

Clinical features including MAPK CTG repeat number (peripheral blood), Mercuri score from conventional MRI (cMRI), fat fraction from quantitative MRI (qMRI), pathology and inflammation scores ... [Method, Mapping the UMAP spectrum with clinical features from different myopathies, Paragraph 1]

e. Repeated measurements in LGMD R12: was the within subject variation modelled? Or where samples considered as independent? Are samples obtained from same cases closer than across individuals? This cannot be appreciated in the figure. It is not clear what is meant by oval distribution at line 191. Does this refer to figure 3c?

Response: The UMAP plot illustrates the similarities among different muscles, with muscles that are more alike in terms of their transcriptome profiles tending to cluster together. In this plot, each dot represents an individual muscle sample.

The genetic data from LGMD R12 muscles, where three biopsies were taken from the semimembranosus, vastus lateralis, and rectus femoris muscles of each patient (resulting in three dots per patient), showed less heterogeneity compared to the CDM and FSHD muscles, where only one biopsy was taken per patient (resulting in one dot per patient). Consequently, the LGMD R12 muscles were more closely grouped and formed a compact oval shape on the UMAP plot, whereas the distributions of CDM and FSHD muscles were more dispersed, resulting in an 'elongated ribbon' appearance. This is consistent with common sense.

We have updated the caption of Fig. 3 for enhanced clarity:

C. Mercuri scores, as well as 10-meter and 6-minute walk test results, were also consistent with disease progression in limb girdle muscular dystrophies R12 (LGMD R12) patients. The

more compact distribution seen in LGMD R12, as compared to CDM and FSHD, might be due to the repeated biopsies taken. For each LGMD R12 patient, three biopsies were collected (represented by three purple dots), whereas for each CDM and FSHD patient, only one biopsy was taken (represented by a single purple dot). [Fig. 3 caption]

f. How was the mapping of clinical and MRI scores to the UMAP done? I understand such data were obtained from supplementary data of original studies. Was it possible to match the data to the individual? Does this perhaps explain the reduction in n compared to the table? If so, I would clarify this and include the clinical data into the integrated data or provide an extra table with such pairing.

Response: Thank you for your suggestions. We acknowledge that some clinical features, such as MRI and pathology scores, are specific to the muscle level, while others, like the 10MWT, 6MWT, and clinical severity score, pertain to the individual level.

As addressed in the response to your point about Figure 3B, certain clinical information was missing for some myopathy patients in their original studies. Regarding the supplementary data from these original studies, we are mindful of the ownership rights of the authors. Therefore, we deemed it inappropriate to re-upload this data as part of our supplementary materials. However, we have made available on our GitHub repository the data that was processed and used in this study (<https://github.com/Hirriririr/Myopathy-Spectrum/blob/main/Validation/Jonckheere%20trend%20test.xlsx>).

g. Would it be possible to provide more insight on how the myopathic signature relates to clinical signature. What genes are contributing the most to the UMAP?

Response: While both UMAP and PCA are dimensional reduction methods, UMAP is distinct in that it does not facilitate the investigation of genes contributing most to diversity. Instead, it primarily reflects the similarity among different data points. Your suggestion is insightful, and we plan to incorporate it into the next phase of our study by employing other analysis methods that can delve deeper into this aspect.

3. Deconvolution. eFigs 2 and 3.

a. Line 207: If I understand well 5 controls per group were chosen for deconvolution. Why choosing a selection and not all? And how were the 5 controls cases chosen for each group? Could you please clarify here and in the manuscript?

Response: In our study, as detailed in Table 1, we had eight control groups from various sources including GTEx, GEO, and Helsinki, with different categories including Control (accident death) n=31; Control (unexpected death) n=203, Control (intermediate death) n=46, Control (ventilator case) n=424, Control (slow death) n=87, Control (others) n=111, Control (amputee) n=24, Control (hyperkalemia) n=3. For the deconvolution analysis, we selected five control groups from GTEx muscles, namely those classified as Control (accident death), Control (unexpected death), Control (intermediate death), Control (ventilator case), and Control (slow death).

There are several reasons: 1) We aimed to investigate deconvolution in relation to the gradual muscle deterioration process. Therefore, the control ranking in GTEx muscles (accident death → unexpected death → intermediate death → ventilator case → slow death) was most suitable for this purpose. 2) Robust sample size was met in the GTEx control groups. 3) GEO controls are sourced from many studies which may introduce considerable 4) The sample size of Helsinki controls was small.

b. I believe proportions of cells are plotted and they are expressed in total %. I would suggest to not underline minor differences of a few % points as these could be more related to sampling rather than biological differences. For example the presence of a blood vessel could skew such proportions.

Response: Thank you for your suggestion. We have included this point in the Results section.

Note that the proportion deconvoluted can be significantly impacted by biopsy selection; therefore, minor differences should be interpreted with caution. [Results, Tissue deconvolution showed shared features among myopathies, Paragraph 1]

c. It is unclear how the comparison was done: myopathy vs rest (healthy controls pooled)? If so, why?

Response: Thank you for raising this point. We did not conduct a pairwise t-test specifically for myopathy versus control groups. Instead, this comparison can be directly observed in the bar plots, where the error bars represent the 95%CI of the proportions. We believe that a direct comparison of predicted proportions might not yield significant insights and could be somewhat unreliable. Furthermore, as you pointed out earlier, the selection of biopsy samples can also influence the results regarding proportions, especially for those cell types that have small percentages.

d. I also wonder whether certain cell types (like CD8) were found to be more present in IBM samples where the inflammatory signature and presence of CD8 cells is expected.

Response: This can be an interesting question. However, our deconvolution analysis was based on models trained using publicly available skeletal muscle single-cell datasets (referenced as 10.1126/science.abl4896 and 10.1186/s13395-020-00236-3). It's important to note that T-cell infiltration is typically minimal in healthy skeletal muscles, usually less than 1%. Consequently, the T-cell numbers in these datasets were quite small. Further subclassifying them into CD4 and CD8 T-cells could introduce additional randomness into the deconvolution results, potentially affecting their reliability.

e. Certain relevant cell types such as fibroadipogenic progenitor cells seem to be totally absent? What is the reason for that?

Response: In our study, we adhered to the cell type classifications as defined in the original single-cell dataset. It's possible that fibroadipogenic progenitor cells were either minimally present or not distinctly classified in the original study, which would explain their apparent absence in our results.

f. To what extent the lack of disease specific single cell dataset affects the deconvolution? Perhaps this comparison could be made for 1 disease for which single cell data are available. I also wonder whether the proportion of cells in the single cell dataset affects the proportions

estimate by TAPE. I would suggest to evaluate this changing the proportions of cells in the single cell datasets by down-sampling cell populations.

Response: Thank you for your suggestions. We think that deconvolution, in the end, is an in-silico prediction, which, while informative, has its limitations. Instead, techniques like flow cytometry or large-scale single-cell analysis might offer more robust insights for this type of research.

In the limitations section of our manuscript, we have included the following point:

Second, sample selection and cell type classifications as defined in the original single-cell dataset may also lead to misinterpretations of the predicted deconvolution results. Therefore, flow cytometry or large-scale single-cell analysis are recommended to address this issue more effectively [Results, Limitation, Paragraph 1].

4. Figure 2.

Line 166 does not seem only right corner, there seems to be a stripe going down showing overlap with donors with general disease. Perhaps related to some non-specific inflammation? or spectrum asymptomatic pre-clinical stages as mentioned at line 177?

Response: Thank you for your advice. We concur that the myopathy muscles situated on the left corner and in the middle of the UMAP plot may represent either non-specific inflammation or asymptomatic pre-clinical stages, yet they still signify some level of muscle damage.

To reflect this more accurately, we have revised the relevant sentence as follows:

It is noteworthy that most myopathy muscles from both GEO (100% sequenced by total RNA) and Helsinki (90.5% sequenced by mRNA) were located at the right corner of the UMAP plot. [Results, A myopathy spectrum revealed after integration, Paragraph 2]

5. Figure 4

a. Fig4a why the numbers of each group are given on top of the figure? Columns identify genes. Shouldn't the n be mentioned on the left for each row?

Response: This is a combination plot. The numbers here represent the sample size of each myopathy when they were sequentially compared with the genuinely healthy control (n = 234). The top five up-regulated genes for each myopathy were listed as columns.

We modify the Fig.4 caption as follows:

A. The top five up-regulated genes in general myopathy and six myopathies with more than 25 muscles are shown. These are compared sequentially with the genuinely healthy control group (n = 234). [Fig.4 caption]

b. Fig4b color use – middle 2 groups difficult to distinguish

Response: The color of these two groups have been changed.

6. Contribution of different muscle types.

a. One of the limitation is the muscle selection. There are 2 studies published where multiple paired muscles were analyzed and sequenced. 10.3389/fgene.2023.1216066 and 10.7554/eLife.80500. Would it be possible to use those datasets to estimate the effect of muscle group and or oxidative/glycolytic muscle?

Response: The two papers you mentioned are really good. We plan to more deeply investigate the myopathy muscles in our next stage of this study, and the date of these two papers can be of great help.

b. Does the data allow to identify genes associated with the patterns of muscles involved in different myopathies?

Response: This is an interesting question. Initially, we also aimed to explore this aspect, which led us to collect biopsy location information for each muscle sample, as detailed in Supplement 1. However, we found that this specific information was not reported in most of the studies we integrated. Consequently, we believe that the currently available data may not be sufficient to conduct a comprehensive analysis of this nature."

7. qPCR: how was the statistics done?

Response: The qPCR results were analyzed using the Mann-Whitney test, as indicated in the bottom right corner of Fig. 4c.

We have also included this information in the caption of Fig. 4c for clarity:

The results of the qPCR analysis were compared using the Mann-Whitney test. [Fig. 4 caption]

8. Discussion

a. The authors mention in the abstracts (lines 34-36) and in the discussion that they bridged the gap between muscle bulk-RNA sequencing and clinical features. I find this to be not represented in the data. The authors show that patients cluster on specific areas based on their gene expression and that relates to some extent in LGMD R12 patients (mostly for the X3 group), however this was not the case for other diseases. A clear linear relationships between gene expression and clinical features was not reported. I therefore suggest amend such sentences.

Response: Thanks for your advice.

This spectrum was further partly validated in three specific myopathies (97 muscles) through clinical features including CTG expansion, MRI fat fraction, pathology, and clinical severity scores. [Abstract]

Firstly, we bridged the gap between muscle bulk-RNA sequencing and the phenotypes associated with muscle deterioration. [Discussion, Paragraph 1]

b. I would also suggest to discuss why healthy pediatric cases clustered with some pathological samples

Response: This has been discussed in several parts of the manuscript. The "healthy" pediatric cases actually resemble other histopathologically normal control muscles obtained from GEO and Helsinki datasets, and they likely originate from subjects who were suspected of having muscle diseases. Being labeled as "histopathologically normal" doesn't necessarily rule out the

possibility of future or undiagnosed muscle diseases. And this could explain why some "healthy" pediatric muscles clustered with pathological samples.

As quoted by the authors: "Pediatric control biopsies were gifted from the University of Iowa (courtesy of Dr. Steve Moore) after being evaluated as histopathologically normal" (10.1093/hmg/ddac254).

The control samples used in the original CDM study consist of histologically normal pediatric muscles (0-9 years old), whereas the GTEx genuine healthy muscles are all >20 years old.¹⁶ [Discussion, Paragraph 3]

After integration, an obvious phenomenon observed in the UMAP plot was that the control muscles used in almost all myopathy studies (GEO and Helsinki) were predominantly located within the moderate and severe muscle wasting stages when compared with the GTEx muscles as a reference (Figure 4B). This is consistent with clinical practice; since doctors most often cannot obtain healthy muscles due to ethical reasons and instead use histologically normal muscles to represent 'control' muscles. Considering these 'control' muscles are not genuinely healthy and may introduce some biases... [Results, Featured transcriptional genes and pathways using unprecedented control healthy muscles, Paragraph 1]

Minor points

9. Line 58 add reference

Response: A reference (10.1038/s41588-022-01037-8) has been added.

10. Line 173-177 the sentence is not clear. Could this be clarified?

Responses: These two sentences has been changed to:

With the detailed classification of control muscles, a potential myopathy spectrum order was reflected: Healthy→Mild disease→Moderate muscle wasting →Severe muscle disease. Interestingly, unlike single-cell analysis where similar cell types cluster together, the distribution of myopathy muscles in our study is not as compact. We observed a ribbon-like intrusion of

myopathy muscles into the healthy group. This pattern is analogous to the asymptomatic pre-clinical stages observed in clinical practice. [Results, Featured transcriptional genes and pathways using unprecedented control healthy muscles, Paragraph 1]

11. Fig 1 CTG expansion – expansion

Response: Modified.

12. Supplement 3 (line 227) seems to be missing

Response: Uploaded.

Response to reviewer #3

Xiao et al. present a very interesting work transcriptional and clinical spectrum of a selection of myopathies. Even though the manuscript is definitely of interest to the field, some pieces of information are missing.

Response: Thanks for your acknowledgement of this work, and we reply to your points as follows.

Line 36: CTG expansion should be at least once clarified with an appropriate gene name.

Response: We have added the gene name (MAPK) as your suggested:

Clinical features including MAPK CTG repeat number (peripheral blood), Mercuri score from conventional MRI (cMRI), fat fraction... [Method, Mapping the UMAP spectrum with clinical features from different myopathies, Paragraph 1]

Line 87 and 147-160: The myopathy diagnoses should be provided in the Materials and Methods section rather than in Results, as the diagnoses were not the result of the presented work. The particular muscles taken for the biopsy should also be presented, at least for control and myopathy subgroups.

Response: Thank you for your advice. Considering that our study is an integration of various data, the Methods section primarily details the data processing techniques, while the Results section encompasses the overall outcomes of this integration. Therefore, we believe it is more appropriate to include the summary of myopathy diagnoses in the Results section.

Regarding the location of muscle biopsies, we have indeed collected this information and detailed it in Supplement 1. Unfortunately, most of the original studies we integrated did not provide this information, making it infeasible for us to offer a summary of the biopsy locations.

Line 103-104: The gene numbers should be precisely described. I presume 9231 genes had a coverage of at least one in every sample. How were 16953 candidate genes selected? I

presume these are either muscle-specific genes or had coverage of at least one in at least one sample. If these are muscle-specific genes, why the expression of almost half of them was undetectable in every sample?

Response: Since we integrated both mRNA and total RNA dataset, we firstly transformed all gene names to ENSG gene id and intersected them (n = 1221) got 16953 candidate genes (mostly coding RNA genes). Then 9231 genes (muscle-specific genes) were subsequently selected as each count was at least one in every sample.

Responses: We integrated both mRNA and total RNA datasets, initially converting all gene names to ENSG gene IDs. Then we intersect all gene sets across all muscles (n = 1221) resulted in 16,953 candidate genes, predominantly coding RNA genes. Finally, we selected 9,231 muscle-specific genes, ensuring that each had a count of at least one in every sample."

We have modified the sentences for more clarity:

Different gene sets across all 1,221 samples were initially intersected, resulting in 16,953 candidate genes. To minimize the impact of genes with low expression, we applied a straightforward yet stringent filtering rule to each sample: counts for muscle-specific genes must exceed 0 in all samples. Following this criterion, 9,231 genes were selected. [Methods, Preprocessing and Integration Analysis, Paragraph 1]

Line 111, or rather 180-182: The authors should explain what was the reason behind the pseudo-time analysis and trajectory prediction. Did you want to check if control muscles are in the process of aging/degeneration/wasting due to age, any other disease, or additional factors?

Response: Although pseudo-time and trajectory analyses are more commonly employed in single-cell studies, they can reflect the progression or tendency of underlying biological processes. We believe this approach is quite analogous to our scenario, where each dot represents not a single cell, but an individual muscle. Our objective is to use these methods for in-silico prediction of muscle deterioration. This approach is useful since the true healthy states of each muscle (or subject) are already known, enabling validation against de facto evidence.

We have modified this part:

This spectrum order was first validated by in-silico analyses (Figure 3A). Both pseudo-time analysis and trajectory prediction algorithms provide similar muscle deterioration transformation to the severity spectrum. [Results, Spectrum order validation with clinical features, Paragraph 1]

Line 124: How many samples were in the single-cell datasets?

Response: The single-cell datasets comprised 2 muscle samples from the dataset referenced as 10.1126/science.abl4896 and 10 samples from the dataset cited as 10.1186/s13395-020-00236-3. These datasets were sequentially utilized to train two deconvolution models.

Line 157: HyperCKemia does not seem an appropriate control as it is usually a sign of muscle damage.

Response: This is a tricky question. Clinically, HyperCKemia can result from both physiological and pathological factors, including heavy exercise, pregnancy, hyperthermia, and thyroid/parathyroid abnormalities (PMID: 15751568). A study from Norway reported persistent hyperCKemia in 1.3% of the normal population (PMID: 21592795). Additionally, it's observed more frequently in Black individuals compared to their European and South Asian counterparts (PMID: 17892987).

Myology specialists tend to diagnose individuals presenting with muscle weakness or atrophy as myopathy patients, especially when supported by genetic and pathological evidence. However, a diagnosis of myopathy based solely on HyperCKemia is often insufficient. These subjects frequently exhibit Variants of Uncertain Significance (VUS) and, upon further investigation (such as pathology and MRI), typically display a healthy control phenotype. Therefore, we believe classifying them as control samples in our study is more appropriate.

Line 173-175: The mild disease and moderate muscle wasting groups should be precisely described. Are these subgroups of myopathy group with lower UMAP scores of clinical features? If not, how were they performing in Pseudotime analysis and trajectory prediction? Alternatively,

are these subgroups of healthy donors, but those with general diseases and wasting diseases respectively? Could all subgroups, from both healthy and myopathic muscle RNA-seq, be presented in such a spectrum? In addition, was there any correlation between pseudo-time analysis or trajectory prediction and the age of the donors?

Response: We classified the mild disease and moderate muscle wasting groups based on the healthy state annotations from the GTEx database, as detailed on Page 12 of the GTEx Clinical Collection Case Report Form (CRF) [[https://biospecimens.cancer.gov/resources/sops/docs/GTEx_SOPs/BBRB-PM-0003-F6%20GTEx%20Clinical%20Collection%20Case%20Report%20Form%20\(CRF\).pdf](https://biospecimens.cancer.gov/resources/sops/docs/GTEx_SOPs/BBRB-PM-0003-F6%20GTEx%20Clinical%20Collection%20Case%20Report%20Form%20(CRF).pdf)]. These classifications are based on established facts regarding the health states of the muscles/individuals.

we re-classified the GTEx muscle donors into three control groups: 1) 234 very healthy donors (fast death of natural causes or sudden unexpected deaths, e.g., car accident or suicide); 2) 470 donors with mild diseases (ill but death was unexpected or ventilator using cases); 3) 87 donors with moderate muscle wasting (slow death after a long illness, e.g., cancer or chronic pulmonary disease). With the detailed classification of control muscles, a potential myopathy spectrum order was reflected: Healthy→Mild disease→Moderate muscle wasting →Severe muscle disease. [Results, A myopathy spectrum revealed after integration, Paragraph 2]

Additionally, our analyses indicate no correlation between pseudo-time analysis or trajectory prediction and the donors' age. Figure 2A also supports this finding, as the distribution of age and gender is random in the UMAP. This observation basically rules out age and gender as main contributing factors to the observed deterioration spectrum.

Line 234-235: The differently-expressed genes on which pathway analysis was performed should be briefly described (an arbitrarily selected number of top-ranked genes? or rather genes below the given FDR threshold) in the manuscript and listed in the supplementary material.

Response: The DEG gene selection criteria has been described here.

For validating the consistency of the integration results to previous studies, we reanalyzed the original and batch-adjusted read counts using the same processing pipeline (EdgeR) and the same criteria of $|\logFC| > 0.5$ and $FDR < 0.05$. [Results, Featured transcriptional genes and pathways using unprecedented control healthy muscles, Paragraph 3]

Line 560: The subtotal of the demographic information (male proportion and age range) should be provided for the controls (8 control groups together) and myopathies (15 myopathy groups).

Response: Thanks for your suggestion. The demographic table of all these groups was detailed in Table 1.

Reviewers' comments:

Reviewer #2 (Remarks to the Author):

The manuscript is improved and most points were clarified. I would like to ask clarifications on a few points

1. Previous 2d. I believe the gene for CTG repeat number should be DMPK instead of MAPK
2. Previous 2g. I understand the UMAP does not facilitate the investigation of genes contributing to the diversity. Could one however make and compare clusters or assess what genes show a linear association with the distribution of UMAP values on x and y axis?
3. Previous 4. Could the approach suggested at the point above be used to answer this point better?

The original point was

Line 166 does not seem only right corner, there seems to be a stripe going down showing overlap with donors with general disease. Perhaps related to some non-specific inflammation? or spectrum asymptomatic pre-clinical stages as mentioned at line 177?

4. Previous 3d-e. It is clear that the deconvolution analysis depends on the single cell data used to deconvolute. I wonder how insightful is this analysis without the proper reference single cell data. Would it be more informative if a reference less method is used? Alternatively there start to appear more single cell data for different diseases that could be used instead of the control cells.

Response to reviewer #2

The manuscript is improved, and most points were clarified. I would like to ask clarifications on a few points.

Response: Thanks for your constructable comments. We address your comments as follows.

1. Previous 2d. I believe the gene for CTG repeat number should be DMPK instead of MAPK

Response: Thank you for bringing this to our attention. It was indeed a typo error. We have corrected the mistake and carefully reviewed the manuscript for any similar errors.

Clinical features including DMPK CTG repeat number (peripheral blood), Mercuri score from conventional MRI (cMRI), fat fraction from quantitative MRI (qMRI), pathology and inflammation scores ... [Method, Mapping the UMAP spectrum with clinical features from different myopathies, Paragraph 1]

2. Previous 2g. I understand the UMAP does not facilitate the investigation of genes contributing to the diversity. Could one however make and compare clusters or assess what genes show a linear association with the distribution of UMAP values on x and y axis?

Response: Thank you for your suggestion, which we found to be quite interesting. Following your recommendation, we have conducted an analysis to compare clusters and assess genes exhibiting a linear association with the distribution of UMAP values along the x and y axes.

The top 5 genes showing the most positive and negative correlations are as follows. It's noteworthy that MGST1, FASN, and CHRNA1 were also identified as the most significantly featured genes for general myopathy in Figure 4.

By UMAP X	Correlation	P	By UMAP Y	Correlation	P
MGST1	0.220757	6.07E-15	MGST1	0.174749	7.88E-10

ADH1B	0.135342	2.07E-06	CHRNA1	0.109619	1.24E-04
FASN	0.130323	4.91E-06	ADH1B	0.094111	9.93E-04
CHRNA1	0.116791	4.30E-05	CDKN1A	0.093193	1.11E-03
CDKN1A	0.108219	1.51E-04	FASN	0.092374	1.23E-03
...
MYOM1	-0.82893	7.390046e-310	HIBADH	-0.841861	0.00E+00
MAPK9	-0.833469	2.533715e-316	CAMK2G	-0.843715	0.00E+00
KIF13A	-0.834101	3.077364e-317	ANKH	-0.848341	0.00E+00
CAMK2G	-0.837101	1.225283e-321	TPM2	-0.851535	0.00E+00
GDE1	-0.841657	0.00E+00	ACTA1	-0.87211	0.00E+00

9231 genes in total

3. Previous 4. Could the approach suggested at the point above be used to answer this point better?

Response: Technically, UMAP does not exhibit a linear relationship, making correlation analysis less suitable for determining associations using UMAP coordinates. Therefore, we believe that the traditional approach of differential expression analysis, as demonstrated in the manuscript, may be more appropriate for identifying featured genes.

We have made the code and results of the UMAP XY location and gene expression correlation analysis available on GitHub for anyone interested in conducting further analysis.

([https://github.com/Hiriririir/Myopathy-](https://github.com/Hiriririir/Myopathy-Spectrum/blob/main/P8%20UMAP%20coordinates%20and%20Expression%20Correlation.ipynb)

[Spectrum/blob/main/P8%20UMAP%20coordinates%20and%20Expression%20Correlation.ipynb](https://github.com/Hiriririir/Myopathy-Spectrum/blob/main/P8%20UMAP%20coordinates%20and%20Expression%20Correlation.ipynb))

4. Line 166 does not seem only right corner, there seems to be a stripe going down showing overlap with donors with general disease. Perhaps related to some non-specific inflammation? or spectrum asymptomatic pre-clinical stages as mentioned at line 177?

Response: This is exactly what we thought.

Interestingly, unlike single-cell analysis where similar cell types cluster together, the distribution of myopathy muscles in our study is not as compact. We observed a ribbon-like intrusion of myopathy muscles into the healthy and general disease group. This pattern is analogous to the asymptomatic pre-clinical stages observed in clinical practice. [Page 8, Results, A myopathy spectrum revealed after integration, Paragraph 2]

5. Previous 3d-e. It is clear that the deconvolution analysis depends on the single cell data used to deconvolute. I wonder how insightful is this analysis without the proper reference single cell data. Would it be more informative if a reference less method is used? Alternatively there start to appear more single cell data for different diseases that could be used instead of the control cells.

Response: We agree with your opinion. According to our understanding, traditional deconvolution tools also use a reference signature matrix to infer tissue composition. One of the most prevalent tools, CIBERSORT (Newman et al., Nat Methods, 2015), is a good example as it was developed based on a reference matrix termed LM22, which was derived from human hematopoietic cells' microarray and RNA-seq data. Next-generation deconvolution tools are prone to using self-defined reference matrices derived from single-cell RNA-seq data to adapt to situations from different tissues to different diseases. The CIBERSORT next version, CIBERSORTx (Newman et al., Nat Biotechnol, 2019), advocates for users to upload their tissue-compatible single-cell expression matrix as a reference (<https://cibersortx.stanford.edu/>).

Different skeletal muscle single-cell references would definitely impact the deconvolution results. That is why we used two different single-cell datasets (Tabula Sapiens dataset and GSE143704), which were the only two public available ScRNA datasets of human skeletal muscle tissue at our search, to compare the results [Page 9, Results, Tissue deconvolution showed shared features among myopathies.]. Similar results were shown by these two datasets (fewer vasculature structures and more adipocytes and fibroblasts in myopathy samples).

As for your alternative suggestion, we have searched the web for potential usage of myopathy single-cell datasets in the GEO database, Single-cell portal, PubMed, and Google. But we found few (no) publicly available ScRNA data of human skeletal muscle tissue from muscle diseases. For example, 10.1038/s42003-022-03938-0 (deposited but restricted access), 10.3389/fcell.2023.1166017 (deposited but restricted access), 10.1016/j.isci.2023.107479 (ScRNA data of specific cell groups), 10.15252/emmm.202217240 (deposited but restricted access), 10.1096/fj.202201949RR (publicly accessible data but from KO mice and not general scRNA sequencing).

Based on these reasons, we think since we have provided both publicly available data and deconvolution scripts, readers may find it easier for them to analyze the data using their own interested reference matrix.

REVIEWERS' COMMENTS:

Reviewer #2 (Remarks to the Author):

The authors answered all my comments.